# Study on the influence of slope shape with numerical calculation models on slope safety during slope excavation

**Yanping Wang**[1]*, **Liangxiao Xiong**[2], **Hanqiang Wang**[3], **Xiangpeng Ji**[3], **Guang Zheng**[4]

**1** Shandong University of Technology, Zibo, China, **2** East China Jiaotong University, Nanchang, China, **3** No.1 Geological Team of Shandong Province Bureau of Geology and Mineral Resources, Jinan, China, **4** State Key Laboratory of Geohazard Prevention and Geoenvironment Protection, Chengdu University of Technology, Chengdu, China

\* wyp@sdut.edu.cn

**Data Availability Statement:** All relevant data are within the paper and its Supporting Information files.

## Abstract

Under the influence of natural weathering and excavation in human engineering, slopes in nature have various slope shapes. In human engineering activities, the stability of different types of slopes formed by manual excavation must be affected by multiple factors such as geologic setting, lithology and environment. Therefore, to understand the impact of slope shape, geologic setting, and other conditions on slope stability of artificial slopes, calculation models for straight slope, concave slope, and convex slope are constructed based on the three slope shape characteristics. By changing the angles of upward and downward slope angles and analysing the parameters of slope shape, joint spacing, and joint angle, discrete element software is used to calculate the slope safety factor. The calculation results show that the slope shape, joint spacing, and joint inclination affect the safety of slopes. In straight slopes with large joint spacing, the smaller the slope angle, the greater the safety factor. However, in the interval of small joint spacing, the safety coefficient of slopes with slight joint inclination has the opposite variation characteristics. When a<90˚, the straight slope has a dominant joint inclination angle that minimises the slope safety factor. In concave slopes, the more concave the slope shape is, the smaller the safety factor is; For concave slopes with small joint spacing, the slope with slight joint inclination has a more significant safety factor; Under the condition of large joint spacing, there is a dominant joint inclination in the joint inclination range of 30˚ to 70˚ for concave slopes, which minimises the safety factor of the slope. In convex slopes, the smaller the joint inclination angle of the slope, the smaller the safety factor of the slope, and the smaller the upslope angle of the slope, the greater the safety factor of the slope.

## 1. Introduction

The artificial slope is the product of human engineering activities, especially in highway and mining engineering; many artificial slopes are formed due to construction or excavation. In

**Funding:** This work was supported by the natural science foundation of Shandong Province, China (Grant No. ZR2021MD011).

**Competing interests:** The authors have declared that no competing interests exist.

engineering activities, human excavation forms slopes of various shapes, such as concave, straight, and convex. The stability of these artificial excavated slopes is closely related to the shape and geologic setting of the slopes. The stability of these artificial excavated slopes is closely associated with the shape and geologic setting of the slopes. The influence of slope shape, geologic setting, and other characteristics on slope stability has been studied by some scholars. D. H. Gray et al. (2013) established conceptual models and mathematical models for different types of slopes. They determined the influence of slope shape on slope stability and rainfall erosion resistance through laboratory tests and field observations. The analysis and observation results show that the concave slope profile is more stable and produces less sediment than the uniform planar slope. These findings are consistent with the conceptual model and the computer simulation results of soil erosion on unstable slopes [1]. Qiu H et al. (2016) applied statistical analysis and a geographic information system to find the quantitative relationship between the size of a loess landslide and local slope height in different types of slope shapes. The analysis shows that loess landslides occur more frequently in the range of 40 ~ 100m; The steep slopes with high pore water pressure are prone to sliding, and about 34.19% of loess landslides occur in these steep slopes. With the increase of local slope height, the average sliding area and volume of loess increase rapidly. Local slope height and shape are the primary factors controlling loess landslides' occurrence and scale distribution [2]. Zhang t et al. (2017) studied the stability control mechanism of the concave slope of a circular landslide. The research results show that the stability of the concave slope rises less with the increase of the slope height compared with the average gradient, and the enhancement effect of the concave slope is only apparent when the slope height is low [3]. Fan C et al. (2019) found that slope morphology and rainfall characteristics significantly affect the hydrological behaviour of slopes when studying the differences in hydrological changes between slope and non-gully slopes. Due to its topographic features, the underground side flow on gully slopes during rainfall is more pronounced than on non-gully slopes. During heavy rainfall activities in landslide-prone areas, gully slopes may be more prone to instability than non-gully slopes [4]. Shiferaw, H.M. (2020) studied the influence of slope height and angle on slope safety factors and failure modes based on the strength reduction analysis method. The results show that decreasing the slope angle almost linearly increases the safety factor while reducing the height increases the safety factor at a parabolic rate [5]. Gallage C et al. (2021) studied the impact of slope gradient on slope stability and developed a validated numerical model using the test results of the physical model of the slope. The research results prove that with the increase of the slope angle, the slope is more likely to collapse suddenly during rainfall. When the slope inclination is more significant than 1.2 times the soil friction angle, the failure is caused by the loss of soil suction; when the slope is less than or equal to 1.2 times the soil friction angle, the failure is caused by the positive pore water pressure at the slope toe [6]. Wang h et al. (2021) studied the hydrological characteristics of unsaturated loess slopes and the failure process. The results showed that the smaller the slope, the higher the average infiltration rate, the greater the depth of the sliding surface, and the farther the sliding distance [7]. However, the smaller the slope, the longer the irrigation time and water volume required for slope failure.

There are many types of research on the influence of geologic setting on slope stability. C. Lo et al. (2014) investigated the deformation characteristics of subsequent slate slopes between Cuifeng and Wuling in Taiwan. The rock and joint's material strength, the erosion ditch's position, and the joint's dip angle are compared. The analysis results show that reducing rock material and joint strength expands the range of slate deformation. The inclination of joints is the most crucial factor of slate deformation [8]. L.R. Alejano et al. (2017) studied the stability of the granite quarry's large angle inclined (reverse) slope. The research results show that joints and spacing are the most relevant parameters for controlling stability [9]. Q. Meng et al. (2020)

proposed an improved CCVT-based columnar joint structure generation method based on in-situ statistical results and a numerical homogenisation method for determining mechanical parameters. They studied the influence of columnar joint structure on the mechanical properties of the rock mass. The research results show that joint density, thickness, coefficient of variation, and filling material properties affect the mechanical parameters of columnar joint rock mass. With the increase of joint density, the elastic modulus decreases, and the Poisson's ratio increases. The elastic modulus decreases exponentially with increased joint thickness, and Poisson's ratio increases linearly. The coefficient of variation also affects the mechanical properties; The elastic modulus decreases with the increase of the coefficient of variation. The elastic modulus reduction parameter of the filling material is exponential with that of the columnar jointed rock mass [10]. D. Shaunik et al. (2020), the bearing capacity of shallow foundation in rock is affected by non-permanence, number of discontinuous joint tips, dip angle, dip direction and spacing of discontinuous surface, shear strength characteristics of discontinuous surface materials, and intact rock strength [11]. S. Ji et al. (2021) studied and established the power-law relationship between joint spacing and layer thickness of sedimentary rocks and its impact on layered rock mechanics [12]. L. Li et al. (2021) showed that in addition to layer thickness, external stress, volume strain, and pore fluid pressure, the mechanical properties of rocks play a vital role in influencing the joint arrangement in natural rocks. The mechanical properties of the rock are closely related to the stability of the slope [13]. Q. Lin et al. (2021) conducted pressure shear tests on jointed rock specimens with round holes. The results show that the strength of the specimens changes in an inverted U shape with the joint inclination, reaching the maximum at 45˚ [14].

Among the above research results, there are relatively few studies on the influence of slope shape on stability. The effect of slope shape and geologic setting on slope stability must be clarified. In the existing studies, there are many research results on the impact of rock mass joints on slope stability, mainly focusing on the influence of joint strength and spacing on rock mass strength. In contrast, the changing trend of slope stability under the two conditions of slope shape and joint occurrence needs further discussion and research. Therefore, based on the above problems, this study uses discrete element software to analyse and calculate the influence of different slope shapes and joint occurrences on slope stability. The slope shape and joint occurrence conditions are computed by the cross-setting method.

## 2. Test model design

Due to the impact of factors such as topography, geologic setting, human engineering activities, and natural weathering, the slope shape of slopes has significant differences in highway and slope engineering. The slope shape is not fixed, which may be planar, concave, or convex. In the rock slope, because the strata in different regions have different geologic setting characteristics, the relationship between the structural plane of rock mass and the slope body is related to both the geologic setting and the road construction direction, so the relationship between the structural plane and the slope surface is random. The inclination direction of the rock mass structure may be the same as that of the slope, or the inclination direction may be opposite, oblique, etc. Any relationship between the rock mass structure characteristics and the slope surface may occur. As we all know, the rock mass structure affects the stability of the slope surface, so the structural plane characteristics and slope shape may be combined into various possibilities, and their strength will inevitably affect each other.

In practical engineering, the occurrence of rock mass structure and the spacing between structural planes are two critical factors affecting slope stability. Therefore, the strength of slopes with different rock mass structure characteristics and slope shapes must be extra. Based

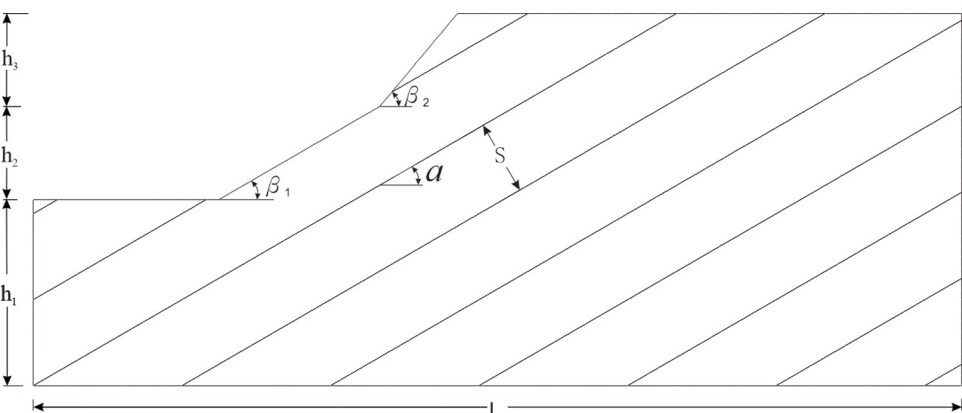

**Fig 1. Abstract model of joint rock slope.**

on the above problems and according to the project's familiar slope shape and rock mass structure characteristics, the experimental model is abstracted (Fig 1).

## 2.1 Numerical calculation model

Discrete element software is used for numerical simulation. As shown in Fig 1, different $\beta_1$ and $\beta_2$ control the slope shape. There are mainly three situations: $\beta_1 = \beta_2$ is the plane slope, $\beta_1 > \beta_2$ is a convex slope, and $\beta_1 < \beta_2$ is the concave slope. The numerical calculation models of the three slope shapes are shown in Fig 2.

Numerical calculation is carried out for the three slope shape calculation models shown in Fig 2. Model boundary conditions: the left and right sides and the bottom are immovably fixed boundaries, and the top is free and unconstrained free boundaries. The physical and mechanical parameters of the model and joint surface are calculated according to the parameters in Table 1.

## 2.2 Numerical calculation group

As shown in Fig 2, the calculation model has different settings $\beta_1$ and $\beta_2$ values to determine the slope shape's degree of concavity and convexity. According to joint inclination $\alpha$, Joint spacing S, and single joint and double joint are calculated by grouping.

## 2.3 Computational constitutive model

UDEC numerical calculation software is used for this calculation. Mohr-Coulomb criteria are the material and joint failure criteria used in the calculation model. UDEC software cannot

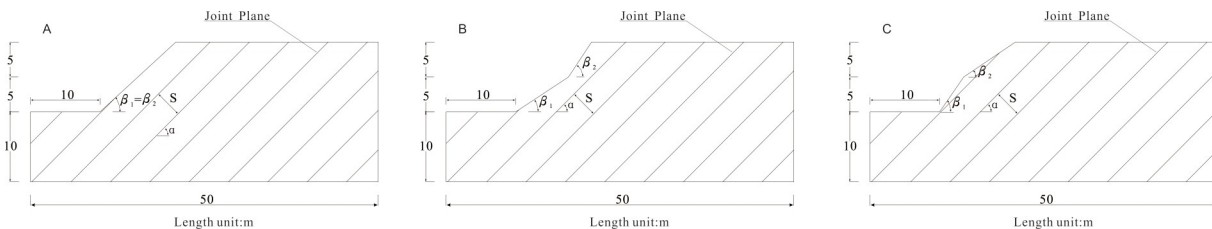

**Fig 2. Numerical calculation model.** A: Flat slope ($\beta_1 = \beta_2$) numerical calculation model B: Concave slope ($\beta_1 < \beta_2$) Numerical calculation model C: Convex slope ($\beta_1 > \beta_2$) Numerical calculation model.

**Table 1. Rock mass material parameters.**

| | Density /kg/m³ | Bulk modulus /GPa | Shear modulus /GPa | Cohesion /MPa | Friction angle /° | Tensile strength /MPa | Dilation angle /° | compressive strength /MPa | Geological Strength Index |
|---|---|---|---|---|---|---|---|---|---|
| Limestone | 2500 | 2 | 2.6 | 4.3 | 31 | 4.3 | 0 | 32 | 55 |
| Joint plane | - | 0.46 | 0.309 | 0.14 | 17 | 0.18 | 0 | - | - |

directly calculate the safety factor. Suppose the safety factor needs to be calculated under the given boundary conditions. In that case, the safety factor can be determined by the load ratio to the design load at the time of failure. As shown in the following formula:

$$F_L = \frac{Bearing\ capacity}{Load} \tag{2.1}$$

The stress state of any element of UDEC can use the principal stress $\sigma_1$ and $\sigma_3$ Expressions. If the minimum primary stress $\sigma_3$ remains unchanged, the maximum principal stress when reaching the limit equilibrium state $\sigma_{1f}$ determined by the following formula:

$$\sigma_{1f} = \sigma_3 tan^2\left(45^0 + \frac{\varphi}{2}\right) + 2c\tan\left(45^0 + \frac{\varphi}{2}\right) \tag{2.2}$$

Where c, φ is the material strength parameter. Then use Formula 2.3 to determine the unit safety factor:

$$F = \frac{\sigma_3 - \sigma_{1f}}{\sigma_3 - \sigma_1} \tag{2.3}$$

## 3. Analysis of calculation results

After defining the calculation model according to the numerical calculation group, assign the physical and mechanical parameters to the model materials and structural planes, calculate the self-stable state of the slope under gravity conditions (by giving the material higher strength parameters), and calculate the stability coefficient of the slope through the limit equilibrium method.

### 3.1 Safety analysis of straight slopes

The straight slope model adopts the calculation group shown in Table 2 and calculates the slope safety coefficient under various parameter combinations. Analyse A, B, and C in Fig 3 and D, E, F, and G in Fig 4 to find that when the joint spacing is greater than or equal to 0.8m, the safety factor of slope decreases with the increase of slope angle. The safety factor of the slope with large joint spacing shows a "^" trend with the change of joint inclination. In sub-diagram A in Fig 4, the joint spacing is 0.2m. The joint angle safety coefficient curve of the slope with a slope of 30˚ is relatively flat, and the curve has the maximum safety coefficient when the joint angle is 90˚. When the slope angle is 50˚ and 70˚, the slope safety factor changes with the

**Table 2. Numerical calculation group with different β₁, α, and S values under β₁ = β₂ condition.**

| Slope angle β₁°) | Joint Spacing S (m) | Joint dip α (˚) |
|---|---|---|
| 30 | 0.2,0.5,0.8,1,2,3 and 4 | 30,50,70, 90,110,130 and 150 |
| 50 | 0.2,0.5,0.8,1,2,3 and 4 | 30,50,70, 90,110,130 and 150 |
| 70 | 0.2,0.5,0.8,1,2,3 and 4 | 30,50,70, 90,110,130 and 150 |

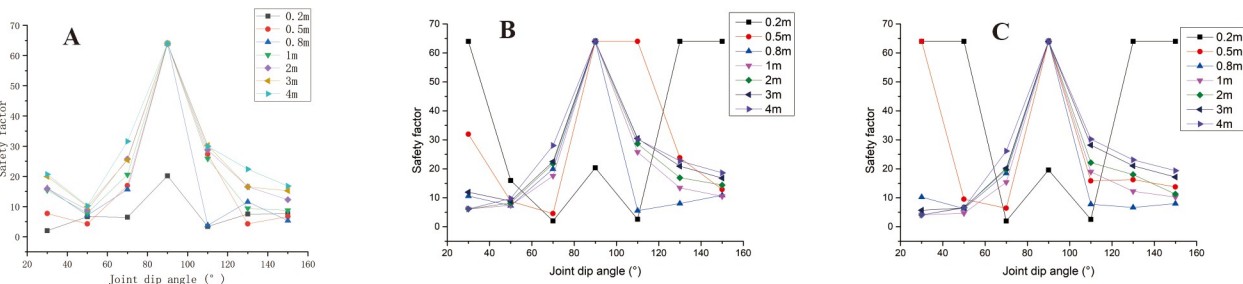

**Fig 3. Curve of joint inclination safety coefficient with different slope angles.** A: The slope angle is 30˚; B: The slope angle is 50˚; C: The slope angle is 70˚.

joint inclination in a W shape. In sub-diagram B in Fig 4, the slope with a joint spacing of 0.5m and slope angle of 30˚ has a similar W shape with the change of joint inclination. The slope safety factor has the minimum safety factor when the slope angle is 70˚ and 130˚, respectively. In the figure, when the slope angle is 50˚ and 70˚, the slope safety factor shows an inverse N-shaped change trend with the change of joint dip angle, and the slope has the minimum safety factor when the joint dip angle is 70˚.

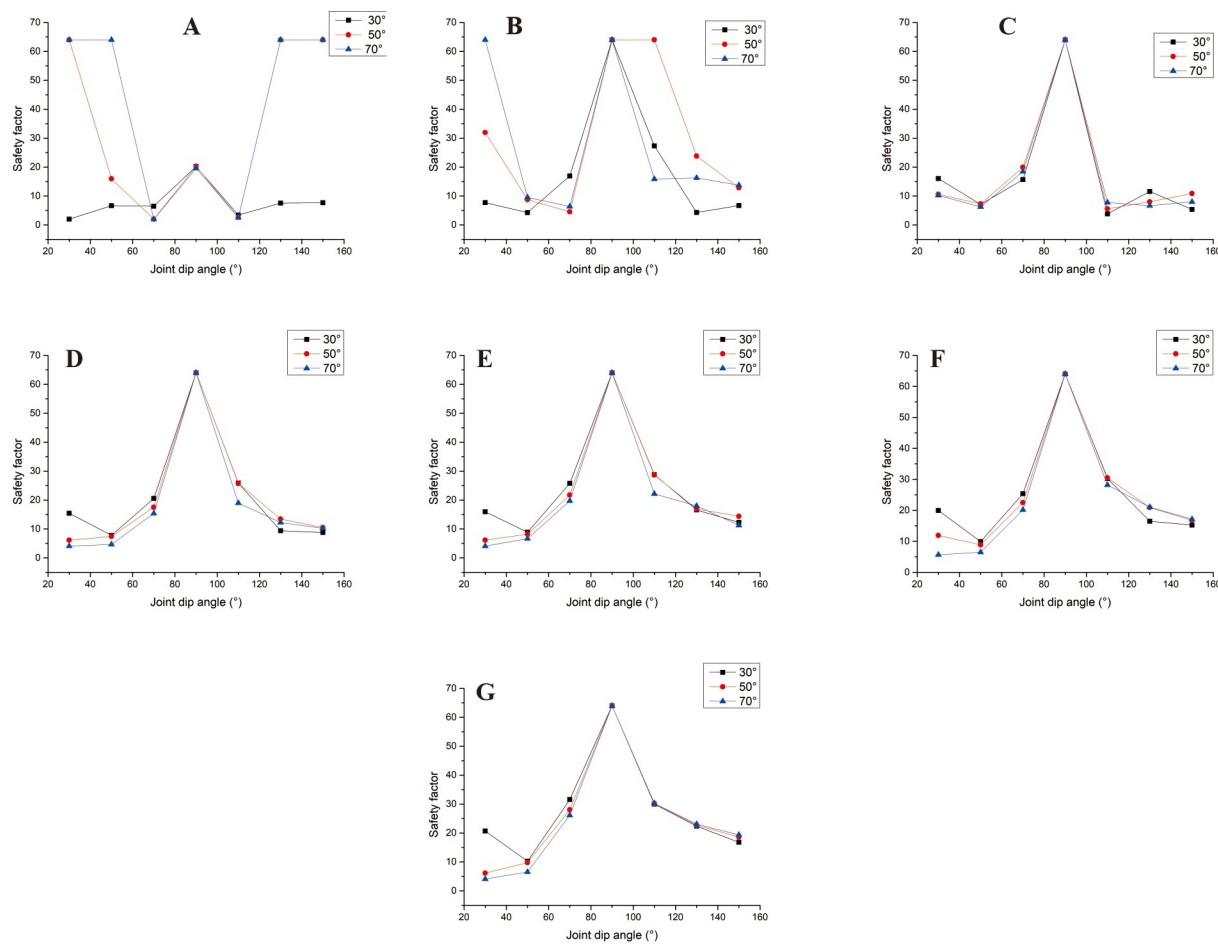

**Fig 4. Curve of joint inclination safety coefficient with different joint spacing.** A. The joint spacing is 0.2m; B. The joint spacing is 0.5m; C. The joint spacing is 0.8m; D. The joint spacing is 1m; E. The joint spacing is 2m; F. The joint spacing is 3m; G. The joint spacing is 4m.

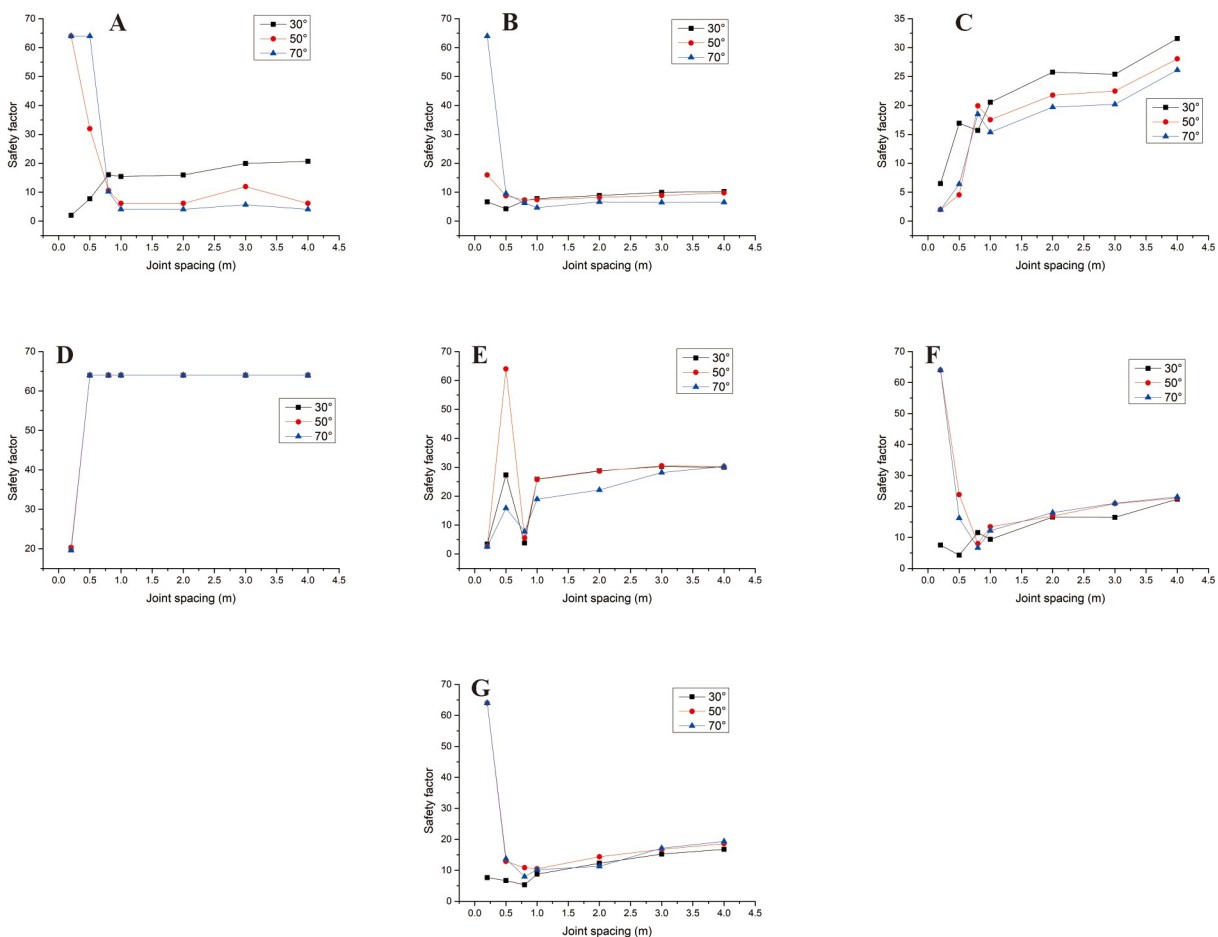

**Fig 5. Curve of joint spacing safety coefficient with different joint dip angles.** A. The joint dip angle is 30˚; B. The joint dip angle is 50˚; C. The joint dip angle is 70˚; D. The joint dip angle is 90˚; E. The joint dip angle is 110˚; F. The joint dip angle is 130˚; G. The joint dip angle is 150˚.

Fig 5 shows the safety coefficient curve of joint spacing for different joint dip and slope angles. In this figure, when the joint dip angles are 30˚, 50˚, 130˚, and 150˚ respectively, the larger the slope angle is, the greater the safety factor is in the interval of small joint spacing (s<0.8m). In the interval of large joint spacing, when the joint dip angles are 30˚ and 50˚, respectively (Fig 5A and 5B), the larger the slope angle of the slope, the smaller the safety factor. The changing trend is opposite to the safety coefficient curve of small joint spacing. When the joint dip angle increases, the safety factor curves of different slope angles gradually approach, which indicates that the impact of slope angle on the safety factor is decreasing. When α = 130˚, α = 130˚, s ≥ 2m (Fig 5F and 5G), the larger the slope angle, the greater the safety factor. When the joint spacing is 0.8m~3m, the influence of slope angle on the safety

**Table 3. Numerical calculation group with different β₁, β₂, α and S values under β₁<β₂ condition.**

| Lower slope angle $\beta_1$ (˚) | Upper slope angle $\beta_2$ (˚) | Joint Spacing S (m) | Joint dip $\alpha$ (˚) |
|---|---|---|---|
| 30 | 50,70 and 90 | 0.2,0.5,0.8,1,2,3 and 4 | 30,50 and 70 |
| 50 | 70 and 90 | 0.2,0.5,0.8,1,2,3 and 4 | 30,50 and 70 |
| 70 | 90 | 0.2,0.5,0.8,1,2,3 and 4 | 30,50 and 70 |

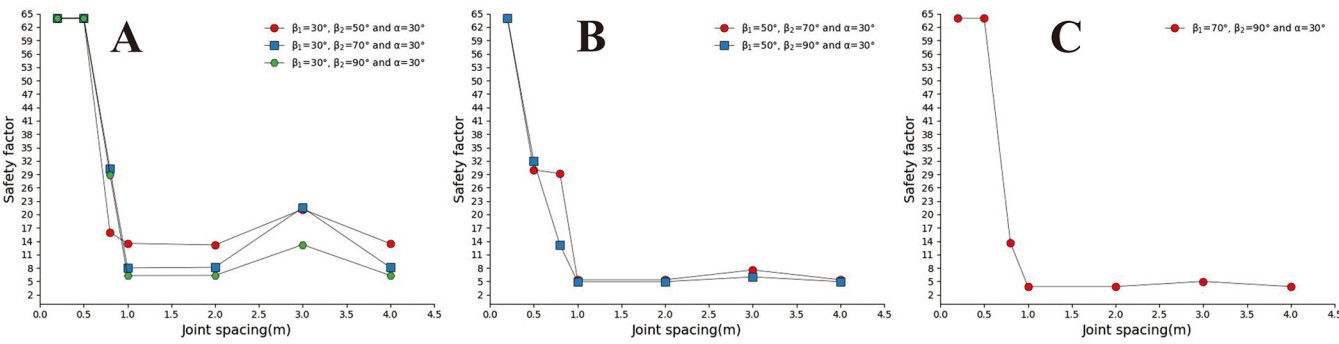

**Fig 6. When α = 30˚, curve of joint spacing safety coefficient for different β₂.** A. β₁ = 30˚; B. β₁ = 50˚; C. β₁ = 70˚.

factor is disordered; when α = 90˚ (Fig 5D), the joint spacing safety coefficient curves of each slope angle coincide. When α = 70˚ (Fig 5C), the safety coefficient curve of joint spacing at each slope angle shows an increasing trend, and only when the joint spacing is 0.8m does the safety coefficient curve fluctuate. When s>0.8m, the safety factor curve shows a trend of smaller safety factors with a larger slope angle. However, when s ≤ 0.8m, the influence of slope angle on the safety factor is disordered. When α = 110˚, s>0.8m (Fig 5E), the safety coefficient curve shows the trend of the more significant the slope, the smaller the safety coefficient, and the safety coefficient curve coincides when the slope angle is 30˚ and 50˚ respectively. When s ≤ 0.8m, the influence of slope on the safety factor is disordered.

## 3.2 Safety analysis of concave slope

A concave slope is a relatively common slope shape in the natural environment, which usually occurs in the mountains with upper complex and lower soft stratum structure. The concave slope is usually composed of two slopes with different slope angles, and the upper slope angle is greater than the lower slope angle, namely: β₁< β₂ (the upper slope angle is β₂. The lower slope angle is β₁). The concave slope model adopts the calculation group shown in Table 3 and calculates the slope safety coefficient under various parameter combinations. In Figs 6–8, when α = 30˚ and α = 50˚, in the small joint spacing interval (0.2m and 0.5m), each type of slope has a significant safety factor, and its overall change trend is that the safety factor decreases with the increase of joint spacing. However, when S>1m, the changing trend of the safety coefficient curve tends to be flat. The above results show that the influence of joint spacing on the safety factor of the concave slope is reduced in the interval of large joint spacing.

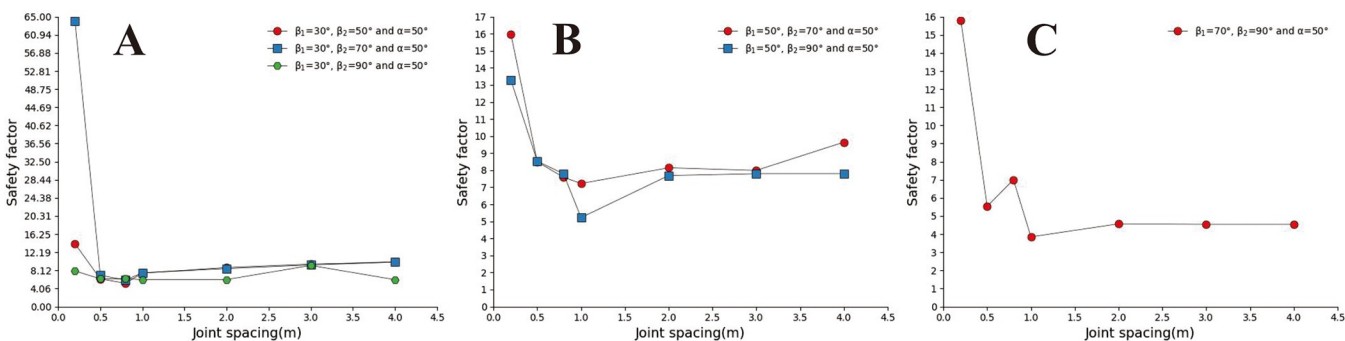

**Fig 7. When α = 50˚, curve of joint spacing safety coefficient for different β₂.** A. β₁ = 30˚; B. β₁ = 50˚; C. β₁ = 70˚.

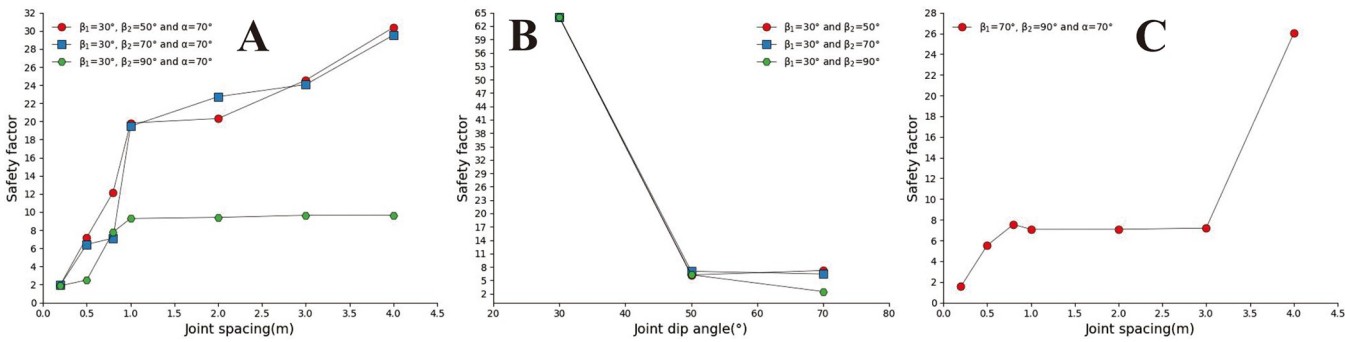

**Fig 8. When α = 70˚, curve of joint spacing safety coefficient for different β₂.** A. β₁ = 30˚; B. β₁ = 50˚; C. β₁ = 70˚.

The changing trend of the safety coefficient curve of sub-graph C in Fig 8 is opposite to that of sub-graph C in Figs 6 and 7. In the three figures, when S ≤ 1m, the safety coefficient curve is steep. However, when S>1m, the safety coefficient curve in the figure changes gently.

Fig 9A and 9B show that the concave slope has a more significant safety factor when the joint inclination is slight under small joint spacing. Under the conditions of large joint spacing and 30˚ ≤ a ≤ 70˚ (Fig 9C–9G), the safety coefficient curve of joint inclination changes in a

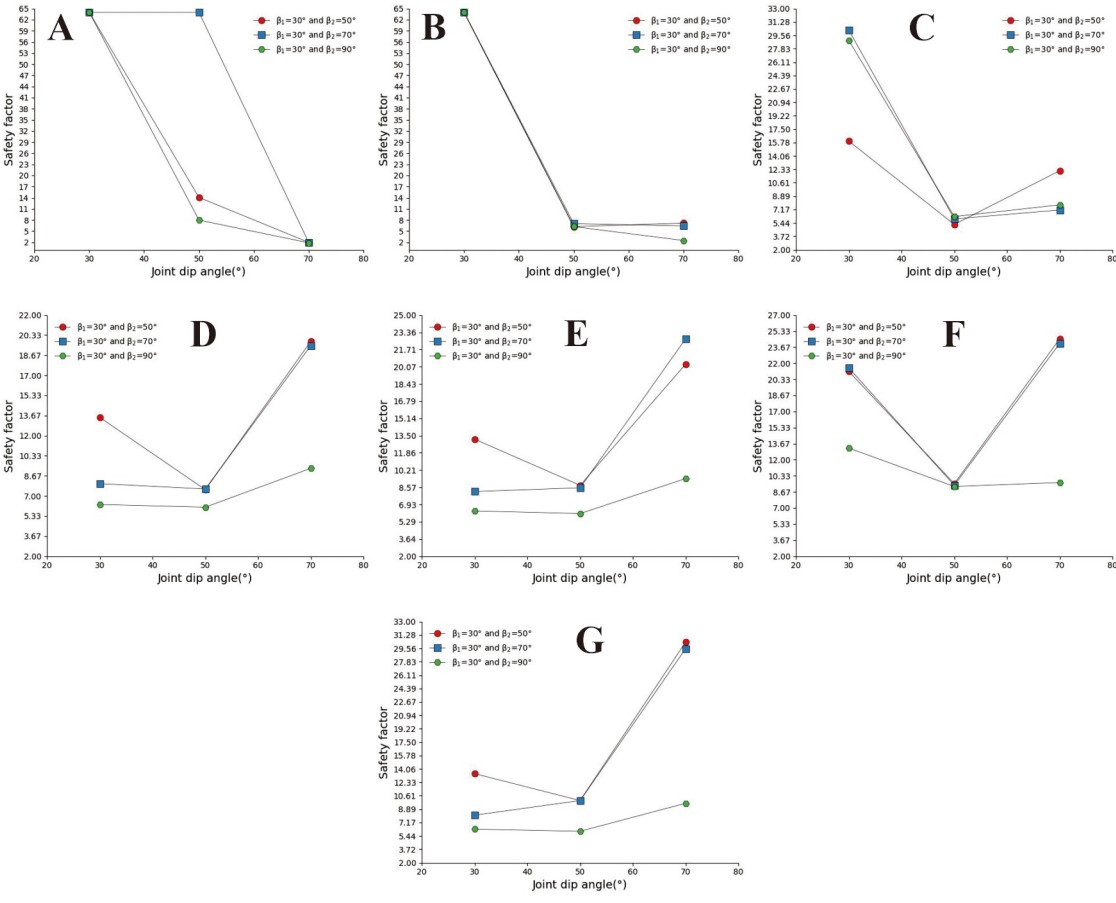

**Fig 9. Safety factor curve of uphill angle with different joint dip angles when β₁ = 30˚.** A. s = 0.2m; B. s = 0.5m; C. s = 0.8m; D. s = 1m; E.s = 2m; F. s = 3m; G. s = 4m.

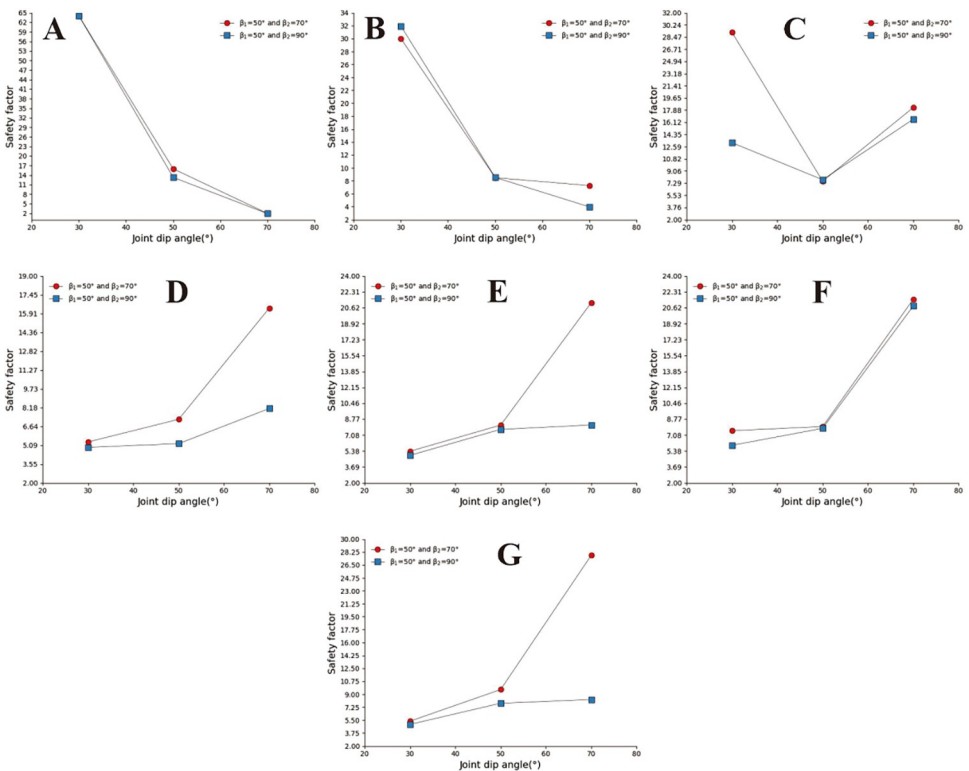

**Fig 10. Safety factor curve of joint dip angle at different uphill angles when $\beta_1 = 50°$.** A. s = 0.2m; B. s = 0.5m; C. s = 0.8m; D. s = 1m; E.s = 2m; F. s = 3m; G. s = 4m.

"V" shape. The above results indicate a dominant joint inclination angle in the concave slope, which gives the slope body the lowest safety factor. In Fig 9, based on the relative position of the safety coefficient curve, it can be seen that when $\beta_1 = 30°$, the larger the slope angle, the lower the slope safety factor. In Figs 10 and 11, under conditions of small joint spacing (Figs 10A, 10B, and 11A, 11B), the safety factor of the slope decreases as the joint inclination increases. When the slope has a slight joint inclination, it often has a more significant safety factor. In Figs 10C and 11C, when s = 0.8m, the joint inclination safety coefficient curve shows a "V" shaped change trend. However, when the rock mass has large joint spacing (sub-graphs D, E, F, and G in Figs 10 and 11), the changing trend of the joint inclination safety coefficient curve shows an increasing trend, which is opposite to the changing direction of the joint inclination safety coefficient curve under minor joint spacing conditions. Fig 10 shows that the safety coefficient curve with a larger uphill angle is generally located below the safety coefficient curve with a smaller uphill angle. The above results indicate that the more concave the slope shape, the smaller the safety factor of the slope.

By analysing Fig 12, it can be found that concave slopes with small joint spacing and slight joint inclination often have more prominent safety factors; In the interval of large joint spacing (S>1m), The safety coefficient curve for slopes with large joint dip angles is generally located above the safety coefficient curve for slopes with small joint dip angles. The above results indicate that in the interval of large joint spacing, the greater the slope joint inclination, the greater the slope safety factor when $\alpha = 30°$ and $\alpha = 50°$, and the safety coefficient curve of joint spacing in each concave slope shows a rapid downward trend in the interval of small joint spacing (0.2m~0.8m). However, when s>0.8m, the joint spacing safety coefficient curve is

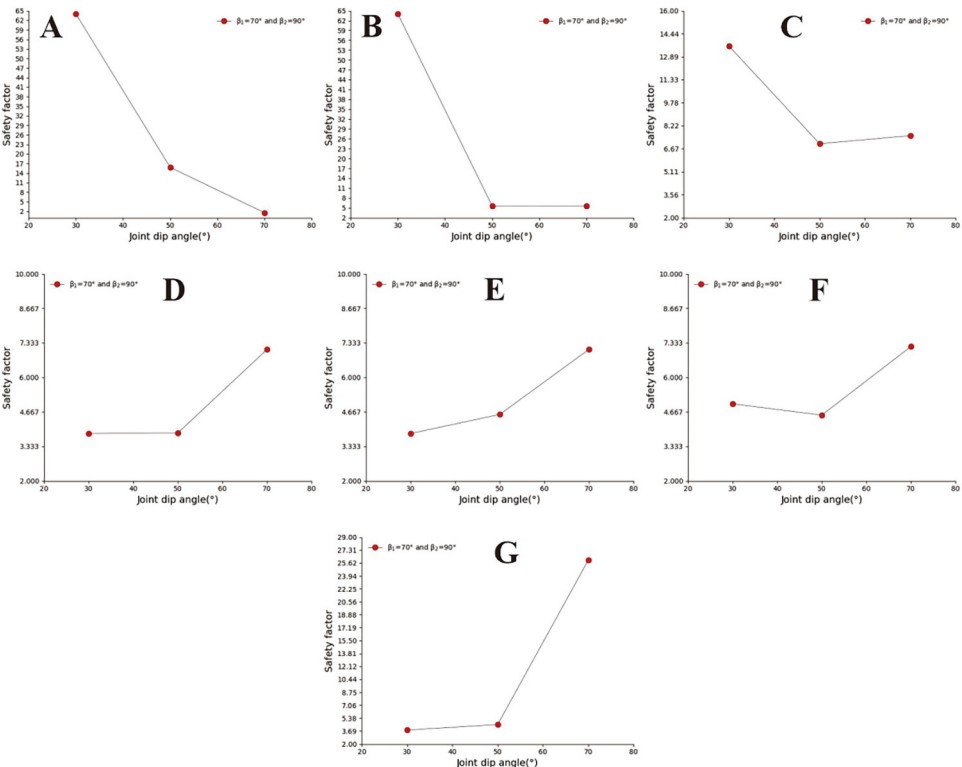

**Fig 11. Joint dip safety factor curve when $\beta_1 = 70°$ and $\beta_2 = 90°$.** A. s = 0.2m; B. s = 0.5m; C. s = 0.8m; D. s = 1m; E. s = 2m; F. s = 3m; G. s = 4m.

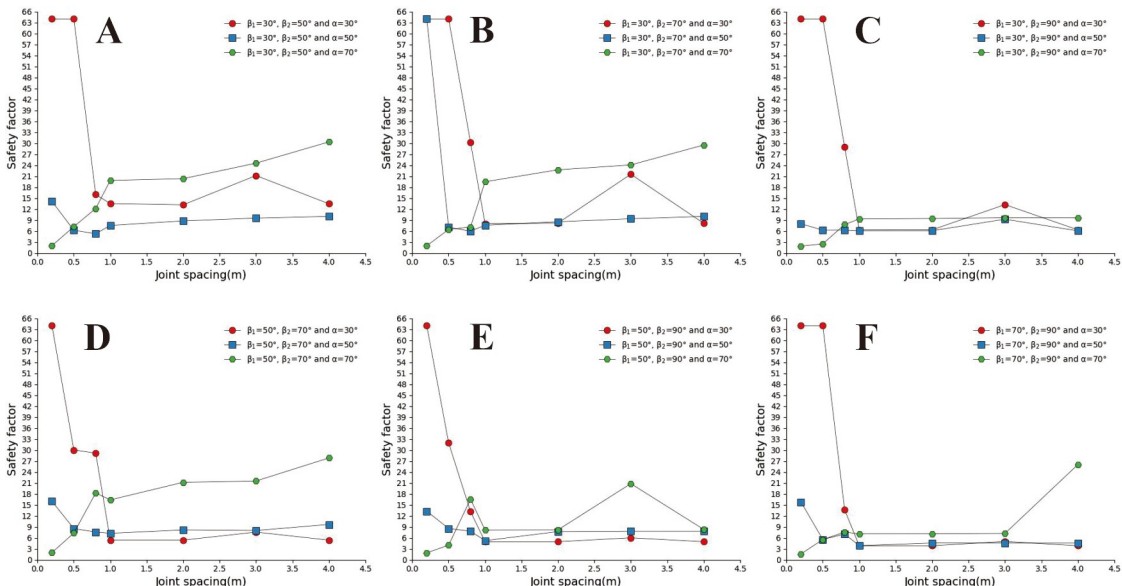

**Fig 12. Safety factor curve of joint spacing with a different joint inclination.** A. $\beta_1 = 30°$ and $\beta_2 = 50°$; B. $\beta_1 = 30°$ and $\beta_2 = 70°$; C. $\beta_1 = 30°$ and $\beta_2 = 90°$; D. $\beta_1 = 50°$ and $\beta_2 = 70°$; E. $\beta_1 = 50°$ and $\beta_2 = 90°$; F. $\beta_1 = 70°$ and $\beta_2 = 90°$.

Table 4. Numerical calculation group with different β₁, β₂, α and S values under β₁>β₂ condition.

| Lower slope angle $\beta_1$ (°) | Upper slope angle $\beta_2$ (°) | Joint Spacing S (m) | Joint dip $\alpha$ (°) |
|---|---|---|---|
| 50,70 and 90 | 30 | 0.2,0.5,0.8,1,2,3 and 4 | 30,50 and 70 |
| 70 and 90 | 50 | 0.2,0.5,0.8,1,2,3 and 4 | 30,50 and 70 |
| 90 | 70 | 0.2,0.5,0.8,1,2,3 and 4 | 30,50 and 70 |

approximately horizontal. This indicates that the influence of joint spacing on the slope safety coefficient is decreasing in large joint spacing intervals, and local fluctuations are related to the cutting position of the joint on the slope surface. When a = 70˚, the slope curve shows a rapid increase trend in the small joint spacing interval and a slow increase trend in the large joint spacing interval.

## 3.3 Safety analysis of convex slope

Convex slopes are commonly found in artificial slopes. Under natural conditions, slopes with this type of slope shape can also be found in slope structures where the upper rock mass is weak and the lower rock mass is hard. The convex slope model adopts the calculation group shown in Table 4 and calculates the slope safety coefficient under various parameter combinations. By analysing subgraph A of Fig 13, it can be found that the safety coefficient of joint

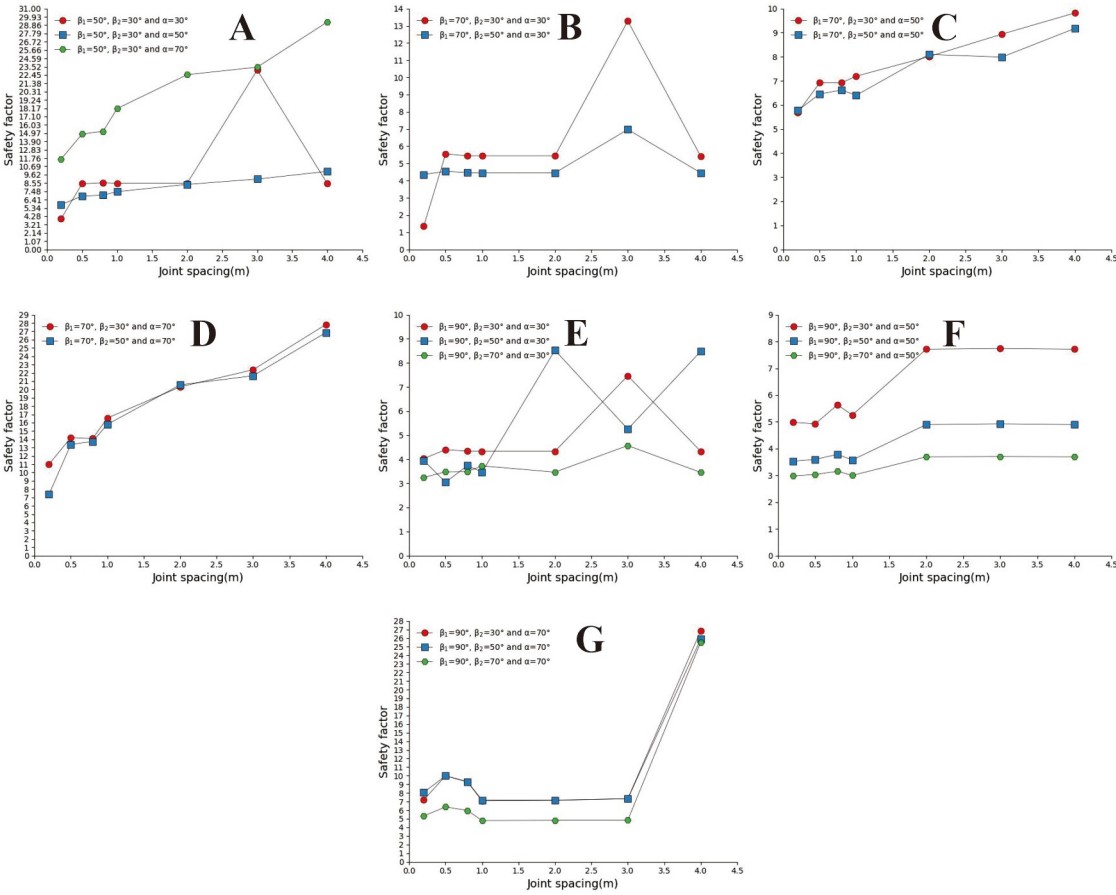

Fig 13. Curve of joint spacing and slope safety factor. A. $\beta_1$ = 50˚ and $\beta_2$ = 30; B. $\beta_1$ = 70˚ and $\alpha$ = 30˚; C. $\beta_1$ = 70˚ and $\alpha$ = 50˚; D. $\beta_1$ = 70˚ and $\alpha$ = 70˚; E. $\beta_1$ = 90˚ and $\alpha$ = 30˚; F. $\beta_1$ = 90˚ and $\alpha$ = 50˚; G. $\beta_1$ = 90˚ and $\alpha$ = 70˚.

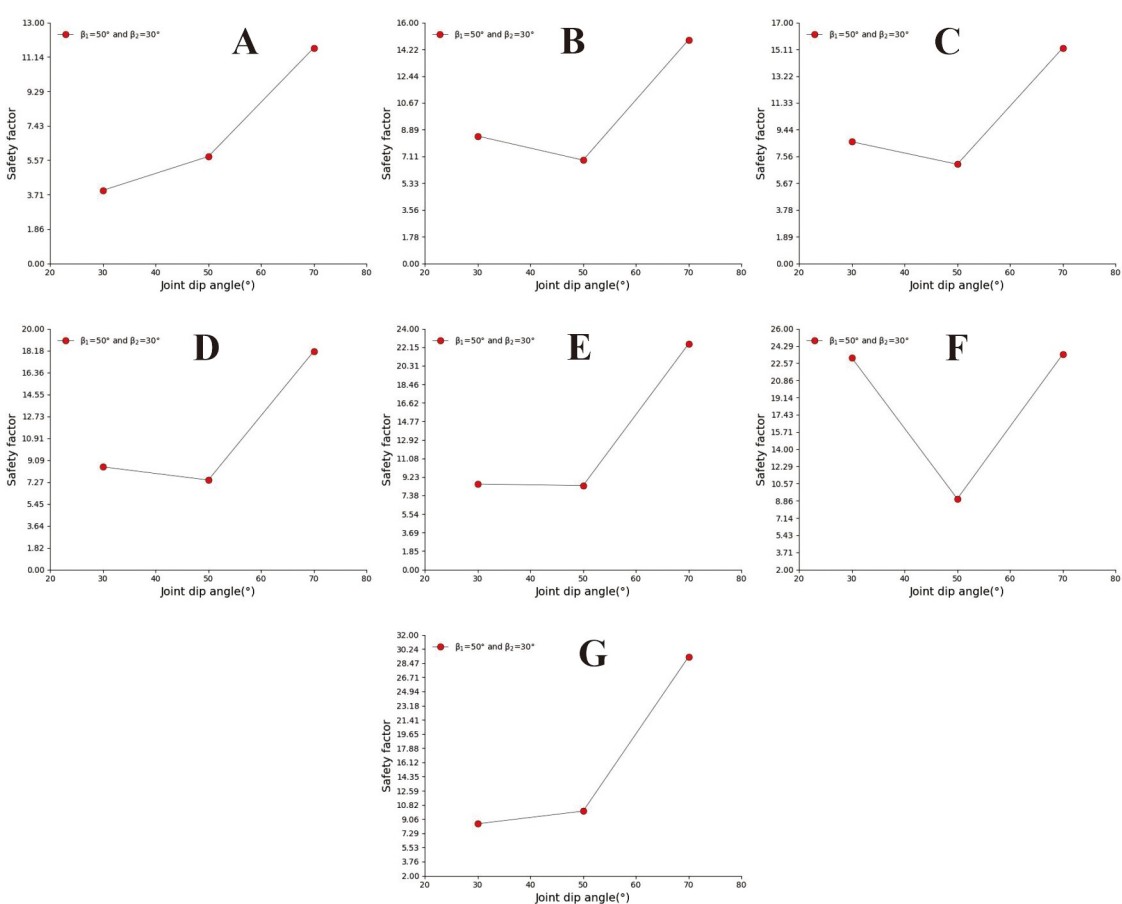

**Fig 14. Safety factor curve of joint inclination with different joint spacing for $\beta_1 = 50°$.** A. s = 0.2m; B. s = 0.5m; C. s = 0.8; D. s = 1m; E. s = 2m; F. s = 3m; G. s = 4m.

spacing presents a curve-increasing trend in convex slopes. In Fig A, when s = 0.2m and s = 0.4m, the greater the joint inclination angle, the greater the safety factor of the slope. When 05m<s<3m, among the three curves with joint inclination angles of 30˚, 50˚, and 70˚, the safety coefficient curve with a joint inclination angle of 50˚ is located at the bottom of the other curve, indicating that the convex slope has an optimal joint inclination angle to minimise the safety coefficient of the slope, analysing the B, C, D, E, F, and G diagrams, it can be found that the safety coefficient curve with a slight uphill angle is usually located above the safety coefficient curve with a sizeable uphill. This indicates that the smaller the upward slope angle of this type of convex slope, the greater the safety factor of the slope.

In Fig 14, when s = 0.2m and s = 4m (Fig A and G), the slope safety coefficient tends to increase with the increase of joint inclination. However, when 0.4m ≤ s ≤ 3m, the joint inclination safety coefficient curve presents a "V" change characteristic. This indicates that the convex slope has a joint inclination angle that minimises the safety factor of the slope. In Fig 15, the safety coefficient curve of joint inclination shows an increasing trend, and only in Fig F, when $\beta_2 = 30°$, the safety coefficient curve presents a "V" shaped change trend. According to the positional relationship of the safety coefficient curve, a safety coefficient curve with an upward slope angle of 50˚ is usually located below the safety curve with an upward slope angle of 30˚. The above results indicate that the larger the upward slope angle, the smaller the safety coefficient of the slope. There is a similar rule in Fig 16.

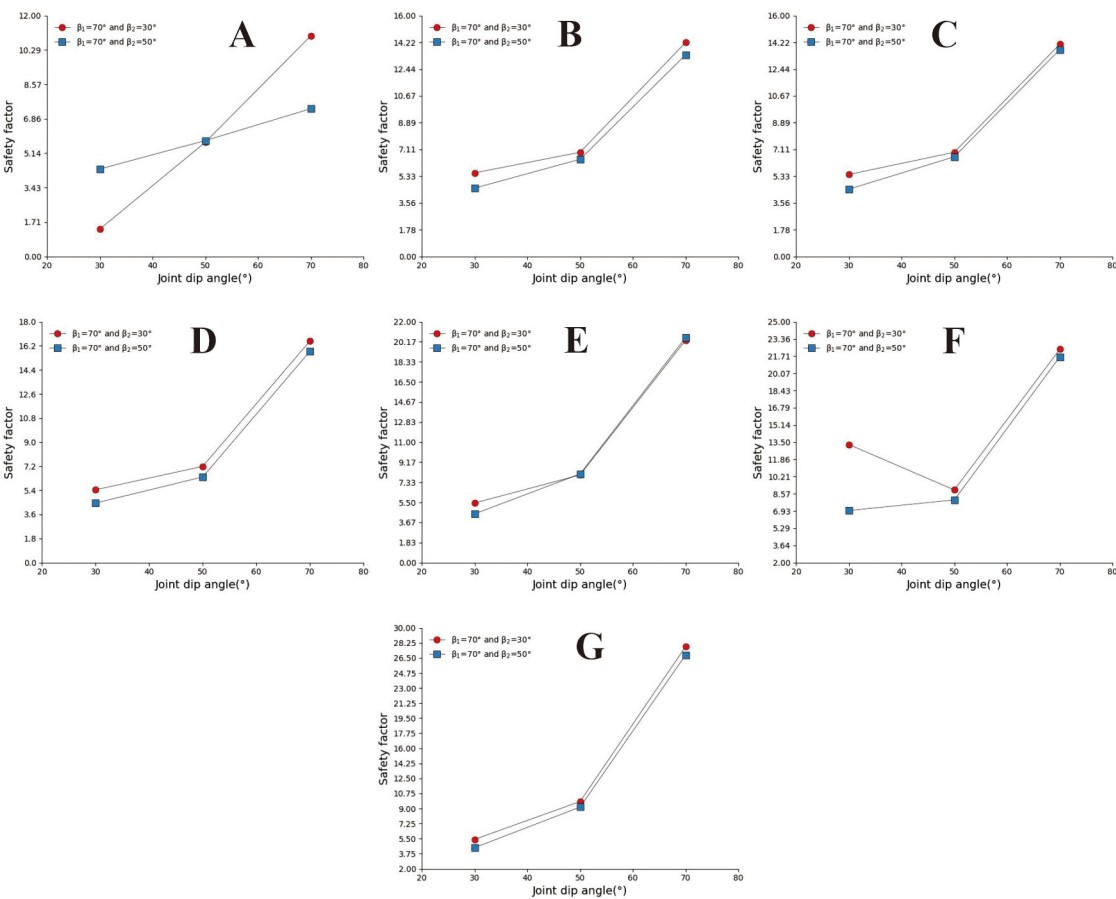

**Fig 15. Safety factor curve of joint inclination with different joint spacing for $\beta_1 = 70°$.** A. s = 0.2m; B. s = 0.5m; C. s = 0.8m; D. s = 1m; E. s = 2m; F. s = 3; G.s = 4m.

In Fig 17A and 17F. The safety coefficient curve with a joint inclination of 50° is located at the bottom, the safety coefficient curve with a joint inclination of 70° is located at the top, and the safety coefficient curve with a joint inclination of 30° is located at the middle. In Fig B, C, and D, the safety coefficient curve for slopes with large joint inclination angles is located above the safety coefficient curve for slopes with small joint inclination angles, indicating that the greater the joint inclination angle of the slope, the greater the safety coefficient. In the D, E, and F diagrams, the downhill angle is 90°. From the range of the safety coefficient curve, it can be found that the safety coefficient curve is located in the low-value area of the ordinate. Only when a = 70 does the safety coefficient suddenly increase.

## 4. Discussion

According to the analysis and calculation results of straight slopes in Section 3.1, the stability of rock slopes is affected by joint spacing and inclination, and their safety factors exhibit a more complex trend of change. As shown in Fig 18A, when s = 0.2m, α = 30°, and α = 50°, the safety coefficient curve tends to increase, but when α = 70°, the change in the slope safety coefficient curve is the opposite. In Fig 18B, when s = 1m, the slope angle safety coefficient curve decreases as the slope angle increases. Gallage C, Abeykoon T, et al. (2021), and Zhang Xing (1988) studied the stability of soil slopes under rainfall conditions through experiments. The research results show that the stability of soil slopes under rainfall conditions will worsen as

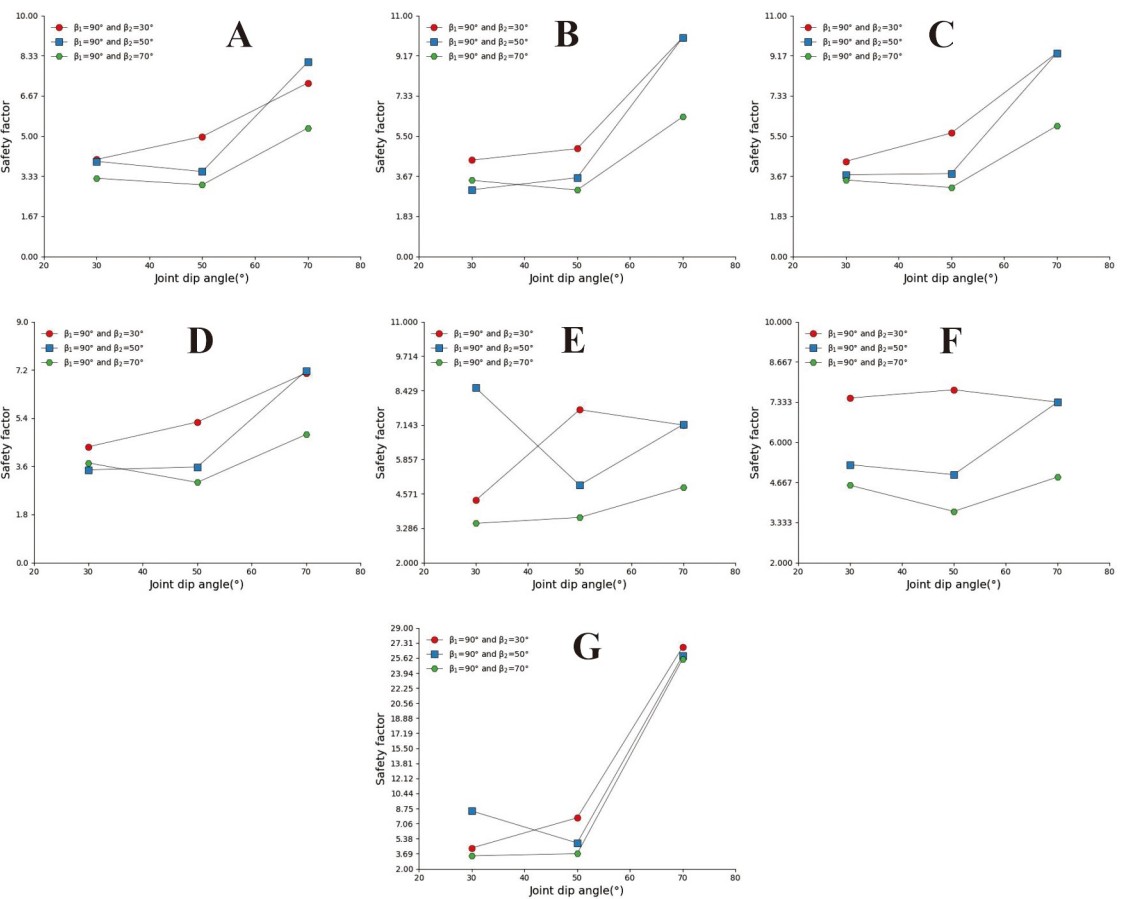

**Fig 16. Safety factor curve of joint dip angle at different uphill angles for $\beta_1 = 90°$.** A. s = 0.2m; B. s = 0.5m; C. s = 0.8m; D. s = 1m; E. s = 2m; F. s = 3m; G. s = 4m.

the slope angle increases. The research results are the same as the variation trend of the safety coefficient shown in Fig 18B. However, in Fig 18A, there is an opposite trend under small joint spacing and inclination conditions.

A concave slope is a slope with a greater upper than a lower slope. According to the calculation results in 3.2, it can be found that the safety factor of concave slopes is also affected by joint spacing and joint inclination, and the safety factor of slopes also presents complex variation characteristics. In Fig 19A, when s = 0.8m and α = 30°, the safety coefficient curve first increases rapidly and decreases slowly as the slope angle increases. When α = 50°, the safety coefficient curve presents a gradually growing trend with the rise of the slope angle. When α = 70°, the safety coefficient curve shows a rapidly decreasing trend and slowly rising as the slope angle increases. In Fig 19B, s = 1m, from which it can be seen that all safety coefficient curves show a downward trend. In Fig A, the variation trends of safety factors with joint inclination angles of 30° and 50° are led by Zhang T, Cai Q et al. (2017) [3], Zhang Xing (1988) [15], and Gray, D H (2013) [1] a similar change rule in the stability study of homogeneous concave slopes. The above results indicate that the more concave the slope shape under large joint spacing, the smaller the safety factor. Under the condition of small joint spacing, the slower the joint inclination, the closer the safety coefficient variation trend is to the safety coefficient variation characteristics of homogeneous concave slopes.

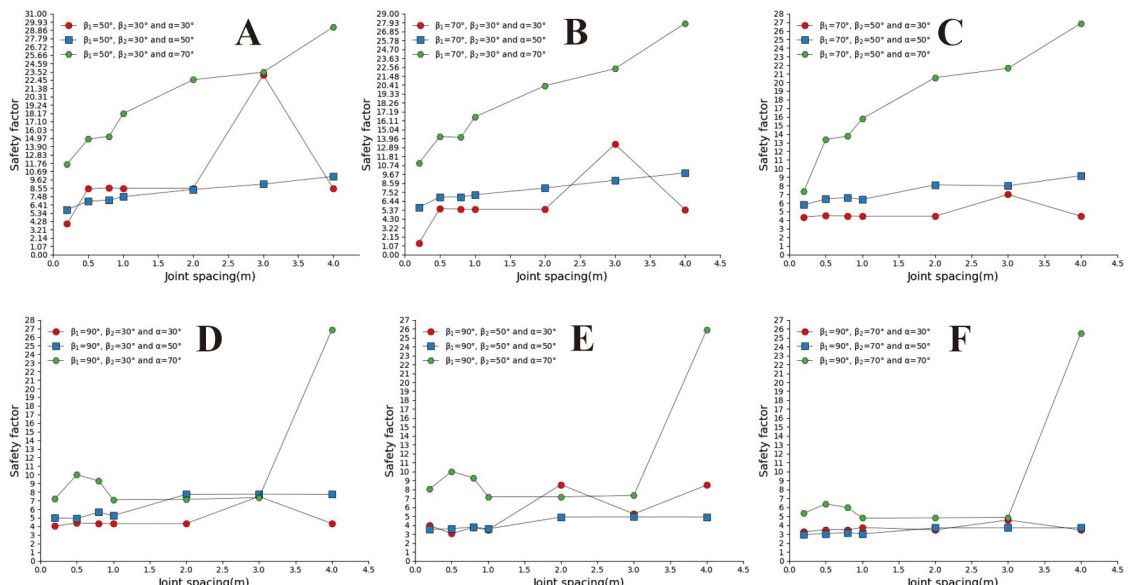

**Fig 17. Safety factor curve of joint spacing with a different joint inclination.** A. $\beta_1 = 50°$ and $\beta_2 = 30°$; B. $\beta_1 = 70°$ and $\beta_2 = 30°$; C. $\beta_1 = 70°$ and $\beta_2 = 50°$; D. $\beta_1 = 90°$ and $\beta_2 = 30°$; E. $\beta_1 = 90°$ and $\beta_2 = 50°$; F. $\beta_1 = 90°$ and $\beta_2 = 70°$.

The convex slope also presents two different slopes up and down, but the lower slope is greater than the upper slope. According to the calculation results in Section 3.3 and the variation trend of the safety coefficient curve shown in Fig 20, it can be seen that joint spacing and inclination impact the safety coefficient of convex slopes. Still, their impact is smaller than that on concave and straight slopes. In Fig 20, the safety coefficient curve shows a downward trend as the downhill angle increases. The above results indicate that the more convex the slope shape, the smaller the safety coefficient of the slope.

What is the difference in safety between straight, concave, and convex slopes? The research results of **Ji F, Shi Y, et al. (2017)** [16] show that the underwater stable slope angle of different slope shapes of soil slopes is as follows: concave slope has the smallest underwater steady slope angle, followed by straight slope, and the convex slope has a relatively large stable slope angle. However, the safety factors for different slope shapes of rock slopes have other magnitude

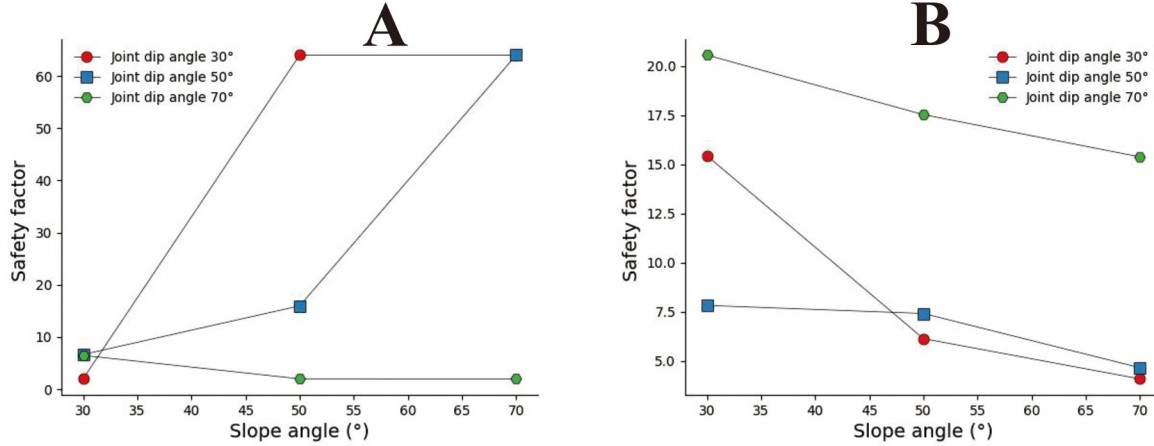

**Fig 18. Slope angle safety coefficient curve of the straight slope with a different joint inclination.** A. S = 0.2m; B. S = 1.0m.

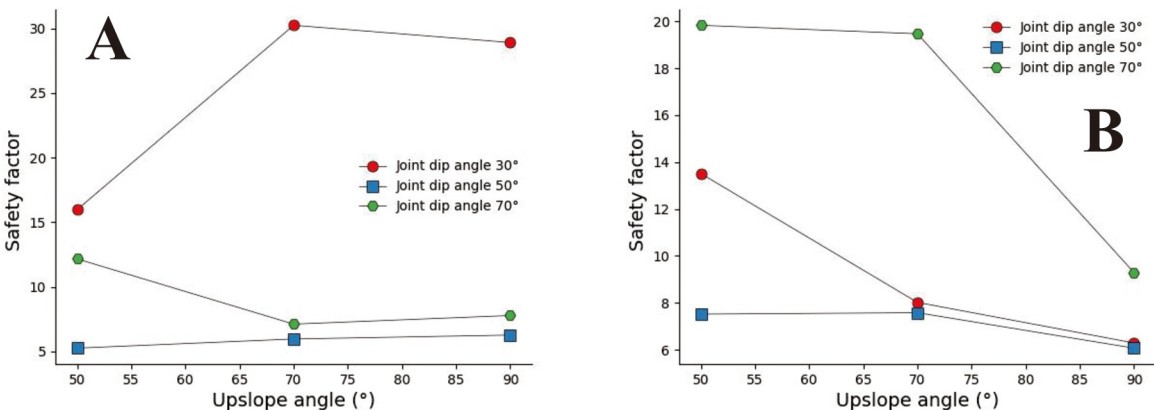

**Fig 19. Slope angle safety coefficient curve of the concave slope with a different joint inclination.** A. $\beta_1 = 30°$, S = 0.8m; B $\beta_1 = 30°$, S = 1.0m.

relationships. In Fig 21 (a = 50°, s = 1m, convex slope $\beta_2 = 30°$, concave slope $\beta_1 = 30°$), it can be seen from the figure that the safety coefficient curve for straight slopes is located below all curves. In contrast, the safety coefficient curve for concave slopes is located above all curves, and the middle curve is the safety coefficient curve for convex slopes. This indicates that the concave slope has the largest safety factor among the three slope types, and the straight slope has the minor safety factor. The safety coefficient curve of the concave slope lies between the two.

## 5. Conclusion

Through the above numerical calculation and analysis, the following conclusions can be drawn:

1. In straight slopes with large joint spacing, the smaller the slope angle, the greater the safety factor. However, in the interval of small joint spacing, slopes with small joint inclination have the opposite variation characteristics; the more significant the slope angle, the greater the safety factor.

2. When the joint dip angle is 30°~90°, there is a dominant joint dip angle in the straight slope, which makes the Factor of safety of the slope minimum; When the joint dip angle is

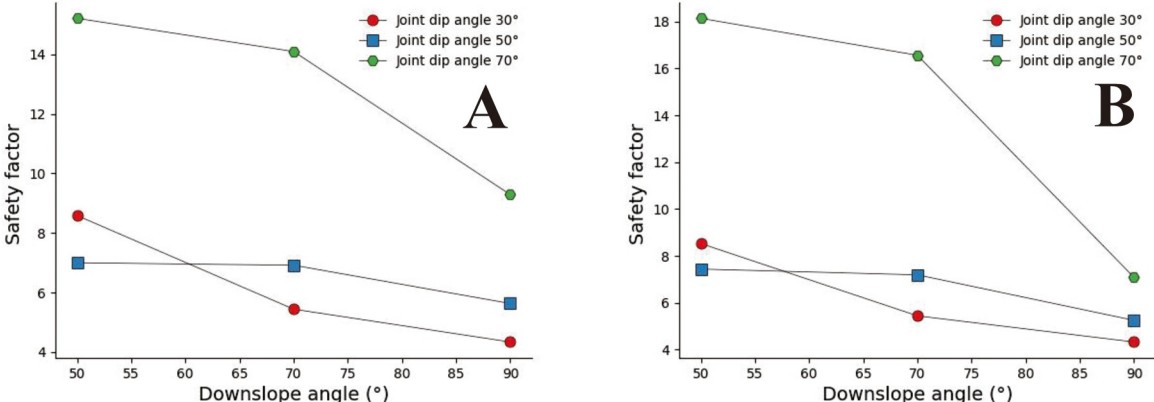

**Fig 20. Slope angle safety coefficient curve of the convex slope with a different joint inclination.** A. $\beta_2 = 30°$, S = 0.8m; B $\beta_2 = 30°$, S = 1.0m.

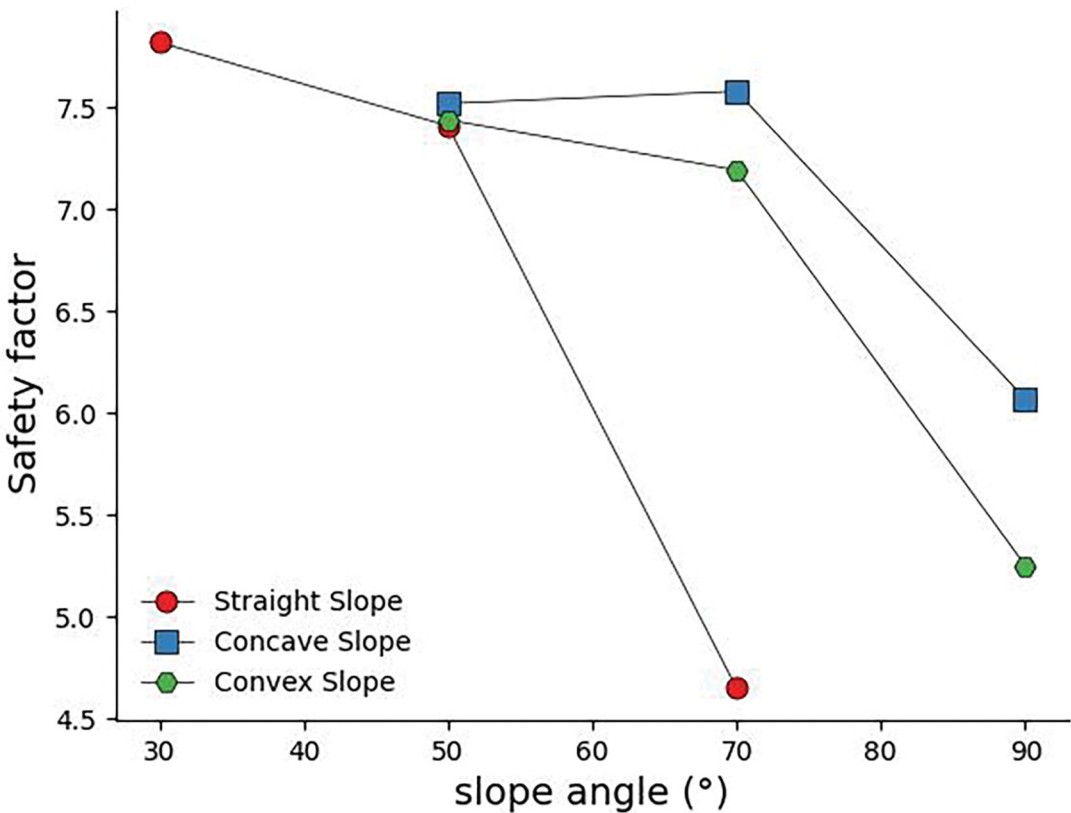

**Fig 21. Slope angle safety coefficient curve of different slope shapes.** S = 1m, α = 50˚.

30˚~90˚, there is a dominant joint dip angle in the straight slope, which makes the Factor of safety of the slope minimum;

3. In concave slopes, the more concave the slope shape is, the smaller the safety factor of the slope.

4. For concave slopes with small joint spacing, the slope with small joint inclination has a more significant safety factor; Under the condition of large joint spacing, the concave slope has a joint dip angle between 30˚ and 70˚ joint dip angles, which makes the Factor of safety of the slope minimum.

5. In convex slopes, the smaller the joint inclination angle of the slope, the smaller the safety factor of the slope. The more convex the slope shape, the smaller the safety factor of the slope.

This study has the following limitations:

1. The calculation model did not consider the non-uniformity of mechanical properties of joints and rock materials. In practical engineering, the mechanical properties of slope joints and rock materials often exhibit certain non-uniformity due to the influence of weathering degree.

2. The calculation model did not consider the impact of multiple sets of joints on slope stability. In practical engineering, slopes often have multiple sets of joints that collectively affect the safety of the slope.

## Author Contributions

**Conceptualization:** Yanping Wang, Liangxiao Xiong.

**Investigation:** Hanqiang Wang, Xiangpeng Ji.

**Methodology:** Yanping Wang.

**Software:** Guang Zheng.

**Writing – original draft:** Yanping Wang.

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
