## [Decision Letter · Decision Letter 0]

18 Jun 2023

PONE-D-23-11131Study on the influence of slope shape and geological structure on slope safety during slope excavationPLOS ONE

Dear Dr. Wang,

Thank you for submitting your manuscript to PLOS ONE. After careful consideration, we feel that it has merit but does not fully meet PLOS ONE’s publication criteria as it currently stands. Therefore, we invite you to submit a revised version of the manuscript that addresses the points raised during the review process.

We look forward to receiving your revised manuscript.

Kind regards,

Abdullah Ekinci, PhD

Academic Editor

PLOS ONE

Journal Requirements:

Reviewers' comments:

Reviewer's Responses to Questions

**Comments to the Author**

1. Is the manuscript technically sound, and do the data support the conclusions?

Reviewer #1: Yes

2. Has the statistical analysis been performed appropriately and rigorously? 

Reviewer #1: Yes

3. Have the authors made all data underlying the findings in their manuscript fully available?

Reviewer #1: Yes

4. Is the manuscript presented in an intelligible fashion and written in standard English?

Reviewer #1: Yes

5. Review Comments to the Author

Reviewer #1: The work submitted to the PLOS ONE entitled as “Study on the influence of slope shape and geological structure on slope safety during slope excavation” is reviewed.

The proposed study examines the to understand the impact of slope shape, geological structure, and other conditions on slope stability of artificial slopes, calculation models for straight slope, concave slope, and convex slope are constructed based on the three slope shape characteristics. By changing the angles of upward and downward slope angles and analyzing the parameters of slope shape, joint spacing, and joint angle, discrete element software is used to calculate the slope safety factor. The reviewer believes this research paper could be interesting to the slope safety during slope excavation those who are interested in slope safety and slope shape models.

In general, a lot of calculation model has been carried out on this paper and the data is adequately analysed however it requires more technical contribution and improvements in the structural layout.

In general, the paper is well structured, and the data is well analysed and requires minor revision to be evaluated. I am suggesting the manuscript to be accepted for publication from the PLOS ONE; however, if the authors are willing to perform minor improvements/corrections on the submitted work.

The article requires English improvement.

Resolution and size of the graphs should be improved for the final submission as they are too small to observe the reported behaviours.

It is quite clear that there is a relationship between the slope angle and slope factor with mechanical parameters such as cohesion and friction angle. Authors can highlight this relationship by using simple statistical tools and developed simple regression models so one can predict one another by simply testing one specimen at certain density. Such approach will increase the novelty of this paper.

6. PLOS authors have the option to publish the peer review history of their article (what does this mean?). If published, this will include your full peer review and any attached files.

Reviewer #1: No

---

## [Author Response · Author response to Decision Letter 0]

6 Sep 2023

Dear Editor and Reviewer:

I have made revisions and responded to each reviewer's comments. The relevant modifications have been marked in red in the "Revised Manuscript with Track Changes" file, and each comment has been replied to in the "Response to Reviewers" file.

Kind regards,

Mr. Wang Yanping, PhD

associate professor，SDUT

Email:wyp@sdut.edu.cn

---

## [Editor Report · Decision Letter 1]

21 Sep 2023

Study on the influence of slope shape with numerical calculation models on slope safety during slope excavation

PONE-D-23-11131R1

Dear Dr. Wang,

We’re pleased to inform you that your manuscript has been judged scientifically suitable for publication and will be formally accepted for publication once it meets all outstanding technical requirements.

Kind regards,

Abdullah Ekinci, PhD

Academic Editor

PLOS ONE

---

## [Editor Report · Acceptance letter]

6 Dec 2023

PONE-D-23-11131R1 

Study on the influence of slope shape with numerical calculation models on slope safety during slope excavation 

Dear Dr. Wang:

I'm pleased to inform you that your manuscript has been deemed suitable for publication in PLOS ONE. Congratulations! Your manuscript is now with our production department. 

Kind regards, 

on behalf of

Assoc. Prof. Dr. Abdullah Ekinci 

Academic Editor

PLOS ONE